# Analyzing Online Reviews to Uncover Customer Satisfaction Factors in Indian Cultural Tourism Destinations

**DOI:** 10.3390/bs13110923

**Published:** 2023-11-13

**Authors:** Aura Lydia Riswanto, Seieun Kim, Hak-Seon Kim

**Affiliations:** 1Department of Global Business, Kyungsung University, Busan 48434, Republic of Korea; auralydia@kyungsung.ac.kr (A.L.R.); ladychulin@kyungsung.ac.kr (S.K.); 2School of Hospitality & Tourism Management, Kyungsung University, Busan 48434, Republic of Korea

**Keywords:** cultural heritage tourism, semantic network analysis, big data analytics, customer experience

## Abstract

Tourism to Indian heritage destinations has been on the rise due to the increasing demand for heritage tourism. Increasing customer satisfaction and promoting Indian culture require tourism businesses to understand factors influencing tourists’ experiences and behavior towards these destinations. Therefore, this study analyzes four popular heritage tourist destinations in India by using online reviews collected from Google Travel. Data are refined, processed, and visualized using the R programming language and UCINET 6.0. Furthermore, we explore the fundamental framework and interconnections among these characteristics through the utilization of exploratory factor analysis and linear regression analysis with the assistance of the SPSS software package. Based on customer reviews obtained from Google Reviews, an analysis was conducted on 6618 reviews of four heritage tourism destinations in India. From the top 60 words, four clusters of words were created, including “Physical characteristic”, “Cultural and historical link”, “atmosphere”, and “area”. Through explanatory factor analysis and linear regression analysis, we found that Physical characteristic, Cultural and historical link, atmosphere, and area all play a significant role in customer satisfaction. This study provides heritage destination managers and Indian government with insights into which attributes impact customer satisfaction the most and offers valuable marketing insights. As a result of this study, we are able to gain a greater understanding of the Indian heritage tourism market, and in doing so, we provide businesses with implications on how to enhance customer service.

## 1. Introduction

The tourism industry has experienced an intense level of competition in recent years, and recent studies on heritage tourism have been refocused from product-centric methodologies to one that focuses on visitors [1]. This new approach aims to investigate the interactions between tourists and heritage sites in the context of this increased competition in tourism. A significant part of the discourse among researchers has revolved around the topic of sustainability, especially in relation to identifying strategies and practices that can facilitate the long-term development of heritage destinations. As part of this research, a number of perspectives will be explored, including economic, social, and cultural perspectives [2].

Due to the growing recognition of sustainable tourism development and the importance of retaining tourists, there has been an increase in interest in studying tourists’ behaviors [3]. Reconsidering and promoting behavioral standards are regarded as a crucial component of the future development of tourism [4]. One of the behaviors of tourists that has gained attention is electronic word-of-mouth (eWOM), which is the sharing of consumption experiences between customers who have purchased, used, or interacted with a product or service [5]. As a result, policymakers, tourism planners, and marketers can focus on key variables that are likely to enhance the tourism experience of their destinations if they implement them correctly [6]. Therefore, this would consequently have a greater impact on encouraging tourists to share information via electronic word-of-mouth (eWOM) in order to share their experiences. In the modern era, word-of-mouth is widely regarded as an important tool for attracting and retaining clients [6].

As a result of the ‘present-centered’ nature of heritage, there is a scarcity of research exploring the sustainability aspect, specifically by examining contemporary communities’ experiences and perceptions of heritage images and narratives through a case study approach [7]. Elevating tourism levels in World Heritage Sites presents a dual outcome, yielding economic benefits while simultaneously posing risks to the sites’ structural integrity, underscoring the need for a comprehensive understanding of the potential advantages and perils following their designation [8].

This research is aimed at exploring the experiences of heritage tourism destinations in India. It employs online reviews as a primary data source to gain insights into customer satisfaction and the influence of specific terminology within the context of heritage tourism, based on word co-occurrence patterns. Consequently, this research endeavors to accomplish the following objectives:Analyze the content and influence of online reviews related to Indian heritage tourism, with a particular focus on frequently employed expressions.Evaluate levels of customer satisfaction when visiting Indian heritage tourism sites.Pinpoint keywords and phrases that contribute to higher review ratings, thereby revealing aspects of the tourism destinations that are highly esteemed by customers and bolstering the business’s reputation in the public eye.

Thus, by using Exploratory Factor Analysis (EFA), this research aims to explore the clusters based on a text mining program [9]. Based on the previous statement above, the purpose of this study is to examine online reviews of heritage tourism destinations and identify significant key influencers within customer networks. This study aims to analyze the relationship between guest feedback and levels of satisfaction, achieved through a rating system. This analysis, in turn, will offer valuable insights into the aspects of evaluation and the factors that exert an influence. Furthermore, the findings from this study provide recommendations for marketing strategies for heritage tourism sites in India that might be useful for the industry in the future. 

This study was organized into several sections. Section 2 presents previous studies and research papers on heritage tourism, online review, and big data analytics. Section 3 outlines the research methodology, and Section 4 provides analytical results with explanatory tables. Lastly, Section 5 includes a discussion of the findings, study limitations, and suggestions for future research.

## 2. Literature Review

### 2.1. Heritage Tourism

Researchers have defined heritage tourism in a variety of ways. As described Poria et al. [10], heritage tourism involves learning about the culture and heritage of a destination through travel. Madden and Shipley [11] stated that heritage tourism is a niche market that is primarily concerned with the preservation of various legacies, including historical buildings. Ensuring the success of heritage tourism hinges on the effective conservation and interpretation of historical sites [12]. Smith and Johnson [12] emphasize the necessity of striking a harmonious equilibrium between preservation and visitor access to guarantee a favorable guest experience. The use of storytelling and interactive exhibits are effective methods for conveying historical narratives and enhancing the engagement of heritage sites [12].

Heritage tourism provides a multifaceted and multidisciplinary perspective on this developing field. It encompasses the fusion of cultural and political influences, highlights the significance of cultural performance, and charts the evolution of research trends within heritage tourism [13]. Furthermore, it accentuates the critical aspect of upholding cultural, economic, and social sustainability [14]. The essential role of cultural performance in heritage tourism is rooted in its ability to bring heritage sites to life, establish connections with the past, educate and engage visitors, and create enduring memories, all while preserving and promoting cultural traditions [15].

The heritage tourism sector is highly profitable, with tourists spending more on their destinations than other types of tourism, and staying longer than regular tourists. There is evidence that heritage tourism is profitable, as heritage tourists tend to be well educated, older, responsible, cosmopolitan, and generous in spending, stay longer, require high-quality services, and are interested in learning more about unique and authentic cultures [16]. The majority of tourism destinations and attractions around the world are focused on elements of cultural heritage, which serves as one of the main drivers of heritage tourism [17]. Consequently, heritage tourism has become one of the most important, widespread, and fastest-growing sectors of the tourism industry [18]. The global heritage tourism market was estimated in 2022 to be worth USD 617.0 billion, and by 2033, the market is projected to reach a total value of USD 1316.4 billion, with a compound annual growth rate of 7.2% [19]. A great deal of this growth can be attributed to the surging popularity of heritage tourism around the world, with India also gaining considerable traction within this sector [20]. Considering these trends, it is essential to improve the experiences and satisfaction of tourists who visit cultural heritage sites, as has been highlighted by Seyfi et al. [21].

### 2.2. Online Reviews and Tourism

A key aspect of tourism and hospitality operations is the integration of customer and supplier systems, which is made possible by modern technology and software applications. Due to the rapid development of technology and the increased use of the Internet, a wealth of digital data, including reviews and ratings published by other tourists, has become more widely available. Travelers often take these sources of information into account when choosing a destination [22]. Several research investigations have highlighted the noteworthy significance of online customer reviews in driving the expansion and advancement of industries and business intelligence [23]. Information and communication technologies are playing an increasingly important role in the tourism industry today, thus making user-generated content essential to Web 2.0. Consumers are involved in the creation of information by creating and sharing user-generated content [24]. The Word-of-Mouth Protocol can be used to promote user-generated content through targeted marketing channels, which is particularly relevant for the tourism sector. Leveraging online word-of-mouth has the potential to decrease the uncertainty and ambiguity that consumers frequently encounter when assessing products and services [25].

In the realm of marketing, UGC platforms can be viewed as a manifestation of consumer-to-consumer e-marketing, akin to the concept of electronic word-of-mouth (WOM) marketing. Here, individuals who hold opinions about a product or service share their perspectives, beliefs, and experiences with others [26]. As highlighted by Fernando (2007) [27], UGC, including social media, stands in sharp contrast to traditional media and marketing methods, as it is the consumers themselves who generate the content rather than the marketers.

The tourism industry is unique among other service industries due to the fact that it involves purchasing and consuming intangible goods. This unique characteristic sets tourism services apart from more traditional services [27]. In today’s market, customers seek accurate and practical information in order to minimize the risk of making a poor purchase. Consumers generally place greater trust in reviews from other customers when purchasing experience-based products, such as travel or entertainment, than when purchasing general merchandise [28]. Research indicates that online reviews frequently offer greater depth and a more precise portrayal of the product or service under scrutiny. Furthermore, these reviews are generally perceived as more dependable than information provided directly by the company [29]. As a result, online reviews play a crucial role in fostering the growth of the tourism industry, offering valuable insights and business intelligence to both businesses and the broader industry [28]. Tourism services are intangible and involve risks, which is why customers rely heavily on the opinions of other tourists when making a purchasing decision.

In the context of WOM’s significant influence on shaping perceptions and decisions, it is crucial to recognize and address potential weaknesses and limitations. Reyes-Menendez, Saura, and Filipe (2019) [30] conduct a deep dive into the challenge of identifying fake online reviews for tourism businesses and the potential harm that counterfeit reviews can inflict on both consumers and businesses. Online reviews, which can mold tourists’ perceptions and behaviors in both positive and negative ways, are not without their drawbacks. These drawbacks encompass issues such as limited information and context, which can potentially lead to misinterpretation [31], an overabundance of repetitive information in reviews that may present challenges in decision making for consumers [32], and the potential to influence overcrowding at popular destinations [33].

### 2.3. Big Data Analytics

As the term “big data analytics” suggests, big data analytics is the process of analyzing vast amounts of structured as well as unstructured data that can be found on the Internet [34]. Sentiment analysis uses big data analytics to identify how users feel about particular products and services based on large online datasets [35]. In recent years, scientists have been able to gather valuable insights from vast amounts of unstructured information online as a result of technological advances [36,37,38]. Especially within the hotel industry, sentiment analysis is gaining traction as a preferred method for examining customer satisfaction data, representing a relatively novel frontier in the realm of data analysis [39].

There has been a long tradition of carrying out customer satisfaction research using surveys and questionnaires as the dominant method of gathering data in the past [40]. Researchers have utilized a range of techniques for conducting sentiment analysis, among them the application of the R language and the RStudio software package [41], RStudio serves as a comprehensive integrated development environment for R programming, offering optimization specifically tailored for statistical computing purposes. UCINET is primarily used in social science research for analyzing network relationships. In the hospitality industry, UCINET is frequently used for sentiment analysis, since it is a reliable and widely used tool [42]. Semantic network analysis delves into the fundamental semantic structure of information by scrutinizing the interconnections between word frequencies and their sequential placement within sentences. In contrast to other forms of analysis, semantic network analysis does not assign specific meaning to the words studied [43].

In a number of previous studies, eigenvector centrality and Freeman-degree centrality have been investigated as two fundamental measures of node power in a network [44]. According to Freeman [45], as opposed to Freeman’s centrality, eigenvector centrality takes into account both the quantity and significance of connections between nodes, whereas Freeman’s centrality is determined by the number of direct connections between nodes. Using CONCOR analysis, we were able to group the most prominent words and extract various features that define the customer experience [44]. A CONCOR analysis uses a co-occurrence matrix to identify hidden subgroups and linkages between them. The identification of key terms defining various dimensions of the customer experience resulted from the application of exploratory factor analysis and linear regression analysis to scrutinize semantic networks.

From the literature review, this study suggested the following hypotheses:

**Hypothesis** **1 ****(H1). **
*Physical characteristic has a positive impact on customer satisfaction in Indian Heritage tourism destinations.*


**Hypothesis** **2** **(H2). **
*Cultural and historical link has a positive impact on customer satisfaction in Indian Heritage tourism destinations.*


**Hypothesis** **3** **(H3). **
*Atmosphere has a positive impact on customer satisfaction in Indian Heritage tourism destinations.*


**Hypothesis** **4** **(H4). **
*Area has a positive impact on customer satisfaction in Indian Heritage tourism destinations.*


Prior research has also identified similar clusters related to customer satisfaction [46], like physical environment [25], atmosphere [36], and area [42]. Simultaneously, the term “Cultural and historical link” pertains to the inherent historical and cultural significance associated with cultural heritage artifacts [46].

## 3. Methodology

A combination of qualitative and quantitative approaches is used to analyze the data in this study to overcome the limitations of traditional survey methods. Walker and Baxter (2019) [47] discussed the interplay between method sequence and dominance, raising questions for researchers, especially those employing sequential mixed methods. This prompts contemplation about the alignment of a quantitative data-dominant approach with the broader research objectives and whether the chosen design inclines researchers toward a particular publication strategy [47]. 

To extract frequently used words from the data, text mining techniques were used in conjunction with Outscraper to facilitate the data collection process. The data were analyzed using Rstudio and a matrix of the most frequently used words was created. These matrices were then analyzed using Freeman’s degree and eigenvector analyses to determine the most significant words in the dataset. Following this analysis, UCINET was used to determine word importance, and CONCOR was used to determine word relationships [45]. Through this approach, valuable insights were gained into how people perceive and understand heritage tourism destinations in India, emphasizing the interconnected nature of these aspects.

Text mining techniques were utilized to preprocess the data, aiming to identify the words that appeared most frequently. A semantic network analysis was then performed on these words to identify those that were most relevant to the research topic [48]. In order to analyze word importance and relationships, CONCOR analysis is conducted with the UCINET. 

Based on the factors from the CONCOR analysis, this study employed quantitative analysis methods. Factor analysis and linear regression analysis are conducted with the SPSS 26, to ascertain the factors reflecting customer experience and their correlation with customer satisfaction. The analysis involved scrutinizing customer satisfaction rating scales, which ranged from 1 to 5, with 1 star representing dissatisfied customers and 5 stars representing satisfied customers. The validity and reliability of the scale were assessed through statistical testing, including the computation of the Kaiser–Meyer–Olkin (KMO) and Bartlett tests, and factor loading higher than 0.5 was considered in conjunction with an exploratory factor analysis [49,50].

The flow of the research is as shown in Figure 1. 

The primary aim of this study was to investigate the determinants impacting customer satisfaction within the context of heritage tourism in India. To acquire relevant information, data were gathered from Google Travel. A list of the four most popular heritage tourism destinations in India was provided on the Ministry of Tourism, Government of India’s website (www.incredibleindia.org, accessed on 15 October 2023), from which data were collected. To ensure that the data were accurate, reviews that contained written comments and a score were retained, while those without written comments were deleted. Further, to ensure the dataset contained recent reviews, only reviews from 2022 and 2023 were collected. As a result of this process, a final dataset of (total number of reviews) was obtained as shown in Table 1.

## 4. Result

### 4.1. Data Pre-Processing

Using a text mining process, customer reviews on heritage tourism destinations in India were collected and analyzed. In total, 7929 reviews were collected and analyzed with a total of 104,523 words. The levels of customer satisfaction were measured using the frequency of numerical ratings, ranging from 1 to 5, as outlined in Table 2. Overall satisfaction was rated at 4.425 out of 5 stars, indicating a high level of satisfaction. The majority of reviewers rated their experiences at 4 or 5 stars, indicating that they were very satisfied with their experience. The remaining 4.2% of customers expressed dissatisfaction with their stay, resulting in a three-star rating. Furthermore, a total of 2.8% of customers provided a low rating of one or two stars, indicating that they were not fully satisfied with their experience. 

The ranking of words based on their frequency in the valid comments was used to determine their significance. Table 3 presents the top 60 words related to the customer experience, which were extracted and arranged accordingly. The selection of these words was based on their significance to the research topic [50]. A proportionate frequency was calculated for each word in the comments using its overall frequency to rank them based on their frequency.

A high frequency of appearance in the customer reviews can be seen in Table 3. Words such as “place”, “tajmahal”, “visit”, “beautiful”, and “nice” were mentioned. In particular, the word “place” was mentioned 2072 times, “tajmahal” 1312 times, and “visit” 987 times. Figure 2 illustrates the interconnectedness between these words through the use of a network containing the most frequently used words. 

Although topic identification can be performed manually from textual data, overlapping words between topics can pose challenges [51]. Moreover, the word segmentation list that was employed fell short in comprehensively encompassing certain dimensions of the customer experience. As an illustration, the high-frequency words underwent semantic network analysis to delve deeper into the relationships and concealed meanings embedded within the text [46]. The purpose of this method was to gain a more comprehensive understanding of the latent meaning conveyed throughout the textual reviews.

### 4.2. Semantic Network Analysis

Semantic network analysis serves as a potent method for revealing implicit connections and meanings among concepts within a specific context. The objective of this method is to examine the links between concepts that are closely related to one another [52]. An analysis of word frequency and clustering can be used by researchers to gain a deeper understanding of the impact of particular words on group relationships and relationships among groups. Through semantic network analysis, concepts are connected, providing valuable insights into textual data and facilitating a more comprehensive understanding of the meanings behind words and connections between them [53]. Studies in this domain have employed metrics such as Freeman’s degree centrality and eigenvector centrality to ascertain the distribution of words based on their proximity to the central points within the network [54].

The Freeman degree centrality of a network indicates the degree of interconnection between nodes, while eigenvector centrality can be employed to identify the most significant nodes in the network [55]. Commencing with the most frequently occurring words, a clustering analysis was executed, coupled with the application of the CONCOR technique. As the name implies, CONCOR stands for “CONvergence of iterated Correlation”, serving as a method to assess the level of similarity between keywords through successive correlation analyses. By analyzing the correlation coefficient between metric values of concurrent keywords, this method can identify blocks of nodes, resulting in clusters containing keywords that are similar to one another [56]. In Table 4, we compare the centralities (Freeman’s degree and eigenvector centrality) to the frequency of the top 60 most commonly used words. The results demonstrate a striking similarity in the distribution patterns of Freeman’s degree centrality and eigenvector centrality, with their values showing a remarkable degree of equivalence.

The findings pointed out that “place”, “tajmahal”, and “visit” exhibited the highest degree of both centrality measures, namely, degree centrality and eigenvector centrality. Compared to frequency measures, the words “India” and “historical” ranked lower in degree centrality, while “good” and “great” ranked lower in eigenvector centrality. Furthermore, “great” and “architecture” were ranked higher than the word frequency for degree centrality, and “historical” and “wonderful” ranked higher for eigenvector centrality. In summary, it is evident that certain words, although less frequently utilized, wielded substantial influence over the network’s structure and impact. Overall, a centrality analysis serves as a valuable tool to identify pivotal nodes within a semantic network, thus facilitating a more precise comprehension of the underlying meanings and relationships.

The implementation of CONCOR-powered semantic network analysis unveiled distinct clusters that mirror reviewers’ experiences with heritage tourism in India. A clustering method was applied to the original reviews taking into consideration the meanings of the keywords, and the resulting clusters were labeled accordingly. Figure 3 illustrates the CONCOR analysis, and Table 5 provides a visual representation of the words belonging to each cluster.

The analysis of online customer reviews led to the identification of four distinct clusters. “Physical characteristic” refer to terms such as “place”, “architecture”, “tomb”, “museum”, “architectural”, “building”, etc. The reviews included the following words: “The architecture of this place is stunning, a true masterpiece of design”, “The place had a certain charm to it due to its unique architectural style and beautiful marble finishes.”, “The tomb was a testament to the incredible architectural achievements of the past.”, etc. In this case, the expressions refer to “Physical characteristic”.

“Cultural and historical link” presents terms such as “historical”, “mughal”, “Indian”, “heritage”, “emperor”, “culture”, etc. The words associated with these terms have historical and cultural connections with India. Reviewers provide several examples, including: “India’s historical heritage is truly remarkable, and the Mughal era plays a significant role in it”, “As an Indian, I take immense pride in the rich culture and heritage of my country”, “India’s heritage is a fascinating blend of different cultures and traditions”, “The Mughal emperors, with their refined taste in architecture and art, left a lasting impact on India’s heritage; The majestic monuments they built are a testament to their vision and grandeur”, etc.

This third cluster includes words such as “nice”, “good”, “great”, “wonderful”, “experience” and more, which relate to the atmosphere of heritage tourism in India. For example, “Exploring India’s heritage tourism sites was a great decision. The atmosphere at the historical monuments and palaces is surreal, and the architecture is simply authentic”, and “I consider my trip to India as one of the most remarkable experiences of my life. The heritage sites were truly captivating, and I was astounded by the immense historical significance they held.”

The last cluster is named “area”, since it contains many words that are related to location in India, such as “visit”, “redfort”, “India”, “agra”, “delhi”, etc. These words also exhibit a relatively high frequency in the context of the reviews (e.g., “tajmahal” was used 312 times, “visit” was used 9875 times). Customer reviews featured the following expressions: “Visiting India was a dream come true for me, and exploring the country’s rich heritage sites like the Red Fort, Agra, and Delhi was truly an amazing experience.”, “The Taj Mahal, in particular, was breathtaking, and walking around its intricate structure and marveling at its beauty was an unforgettable moment.”, “Although the sites can be quite far apart and require a long walk or ride to reach, the journey is definitely worth it.”, etc.

### 4.3. Factor Analysis

Factor analysis, a statistical technique, is employed to explore the variations among keywords within online reviews, aiming to unveil latent patterns. By utilizing oblique rotation, this method facilitates the reduction of numerous variables into more manageable factors. Factor loading is extracted using commonly accepted factorial criteria, with a minimum factor loading of 0.400 used in the final model. Moreover, it is essential that factors possess eigenvalues exceeding 1.0 and account for a substantial portion of the variance. In line with this study’s findings, a total of 18 keywords were categorized into four factors, collectively explaining a significant 93.459% of the variance. The factors identified as independent variables were subsequently employed to pinpoint the critical elements influencing customer satisfaction. As shown in Table 6, the KMO index was 0.826. Bartlett’s sphericity test resulted in a χ2 of 1225.834, with a *p*-value of 0.001, meaning that factor analysis was an appropriate technique for this study.

### 4.4. Linear Regression Analysis

Following the completion of the factor analysis, a linear regression analysis was carried out to ascertain the correlation between guest experiences and satisfaction. A summary of the results is presented in Table 7. The linear regression analysis involves four independent variables, specifically labeled as Physical characteristic (PC), Cultural and historical link(CH), atmosphere (At), and area (Ar). The four variables explained 43% of the variance (R^2^ = 0.430). All the variables have a positive impact on customer satisfaction. “Physical characteristic” (ß = 0.340, *p* < 0.001), “Cultural and historical link” (ß = 0.147, *p* = 0.04), “atmosphere” (ß = 0.291, *p* < 0.001) and “area” (ß = 0.119, *p* = 0.08), as indicated by reviews such as these: “If you’re looking for a travel destination that’s steeped in history and culture, then India is an excellent choice.”; “Indian heritage tourism offers a great experience for anyone interested in history, culture, and architecture. With its impressive buildings, majestic mausoleums, and fascinating museums, India has something to offer everyone.”; “With breathtaking architecture, stunning tombs, museums, and buildings that reflect India’s rich historical past, I highly recommend India to anyone looking for an unforgettable travel experience.” By highlighting the positive aspects of Physical characteristic, Cultural and historical link, atmosphere, and area, guests are more inclined to trust Indian heritage tourism.

## 5. Discussion

India’s rich cultural and historical heritage contributes significantly to heritage tourism in the country. Nevertheless, the sector grapples with a multitude of challenges, including issues of overcrowding, harm to heritage sites, and the looming threat of diminished authenticity. A notable study exploring the repercussions of tourism on heritage sites in India has highlighted the dual-edged sword of this influence. A study focused on the influence of tourism on heritage sites in India reported that while tourism had positive economic implications, it also led to environmental degradation and overcrowding [57]. The Indian government, cognizant of these challenges, actively promotes responsible tourism practices, advocating for the mitigation of negative environmental impacts and the enhancement of cultural sensitivity to nurture sustainable tourism while safeguarding the cultural heritage [58]. An essential facet of promoting heritage tourism is the emphasis on conservation and management initiatives, encompassing preservation, restoration, and rehabilitation of these invaluable heritage sites [59].

The research conducted by Asmelash and Kumar [60], which emphasizes the importance of cultural and environmental preservation, holds the potential to shed light on the vital need to strike a delicate balance between the development of tourism and the conservation of cultural and environmental resources. In the context of clustering, Tao’s research [25] employed a similar clustering approach for the physical environment. However, in their study, it incorporated location-related terminology. In contrast, Handani’s study [42] utilized the “area” cluster in alignment with our research.

Significant research has been undertaken in the domain of heritage tourism. Nevertheless, there exists no prior example of research concerning heritage tourism and visitor contentment in India employing big data. The objective of this investigation was to explore the diverse elements influencing the visitor’s encounter and establish a connection between the visitor experience and their satisfaction. The data resources chosen for this investigation encompassed heritage tourism locations, and online feedback from customers about these locations was obtained as a component of the research procedure. Moreover, a combination of text analysis, semantic network assessment, and quantitative analytical approaches was utilized to unveil hidden meanings and connections within the collected feedback.

## 6. Conclusions

In conclusion, this research underscores the pivotal role played by India’s rich cultural and historical heritage in the realm of heritage tourism. It does not shy away from acknowledging the challenges confronting the industry, the nuanced impacts of tourism, and the government’s proactive stance in promoting responsible tourism. Leveraging a sophisticated methodology, which encompassed semantic network analysis and centrality measures, the study unveiled critical variables that directly influence customer satisfaction. The imperative of striking a harmonious balance between economic development and cultural preservation is a key takeaway, alongside the vital roles played by local communities and focused conservation efforts. Ultimately, this research offers invaluable insights for both academic contemplation and practical implementation within the domain of Indian heritage tourism.

This study harnessed semantic network analysis, employing a dataset of 60 words, to assess centrality measures. Notably, the analysis incorporated Freeman’s degree centrality, eigenvector centrality, proximity analysis, and CONCOR analysis on substantial data. The research was conducted using UCINET 6.0, which includes Netdraw as part of its suite. It specifically honed in on four eminent heritage tourism destinations in India. The analysis conducted through CONCOR identified distinct clusters in users’ comprehension and awareness of the Internet, categorizing them into “Physical characteristic”, “Cultural and historical link”, “atmosphere”, and “area.” Following a factor analysis, 18 words were grouped into four distinct categories: Physical characteristic (PC), Cultural and historical link (CH), atmosphere (At), and area (Ar). It was found that these variables significantly influence customer satisfaction, with each category’s respective beta coefficient and *p*-value indicating the strength of their influence. Collectively, these variables account for 43% of the variance in customer satisfaction (R^2^ = 0.430). In terms of average guest satisfaction ratings, the factors of “Physical characteristic”, “Cultural and historical link”, “atmosphere”, and “area” were found to be statistically significant at the 0.01 level, and they were positively influenced by their standardized coefficient values. The results of this research paper suggest that all hypotheses were validated by the collected data.

The application of semantic network analysis, Freeman’s degree centrality, eigenvector centrality, and CONCOR analysis on big data in this study contributes significantly to the methodological advancements in heritage tourism research. By employing UCINET 6.0 and Netdraw, the research showcases the efficacy of integrating advanced computational tools in studying complex phenomena. The categorization of users’ perceptions into distinct clusters, namely “Physical characteristic”, “Cultural and historical link”, “atmosphere”, and “area”, underscores the multidimensionality of heritage tourism experiences. Additionally, the identification of specific variables, such as Physical characteristic, Cultural and historical link, atmosphere, and area, provides a nuanced understanding of the factors influencing customer satisfaction. These results support the previous studies [61,62]. 

This research holds valuable implications for stakeholders in the heritage tourism industry. By recognizing the pivotal role of distinct experiential dimensions, including Physical characteristic, Cultural and historical link, atmosphere, and area, practitioners can tailor their offerings to align with visitor preferences. Focusing on enhancing these aspects can significantly elevate customer satisfaction levels. Moreover, the integration of advanced analytical techniques like semantic network analysis and CONCOR analysis highlights the potential for leveraging big data in decision-making processes within the tourism sector. By harnessing these tools, policymakers, and industry professionals can gain deeper insights into user perceptions and preferences, enabling them to craft more targeted and effective strategies for heritage tourism development and management.

This study comes with several limitations and these limitations also offer valuable insights for future research directions. The online reviews were sourced from Google Maps, which is the largest search engine globally. However, it is plausible that relying on a single online platform may not encompass all customer preferences. Therefore, for enhanced representativeness, future studies should encompass analyses from various websites. Furthermore, it is worth noting that this study specifically collected data from four heritage destinations. Subsequent research endeavors should aim to encompass a broader array of heritage tourism destinations for a more comprehensive understanding.

## Figures and Tables

**Figure 1 behavsci-13-00923-f001:**
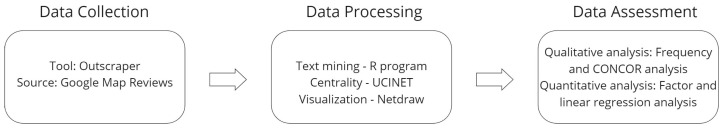
Research flow.

**Figure 2 behavsci-13-00923-f002:**
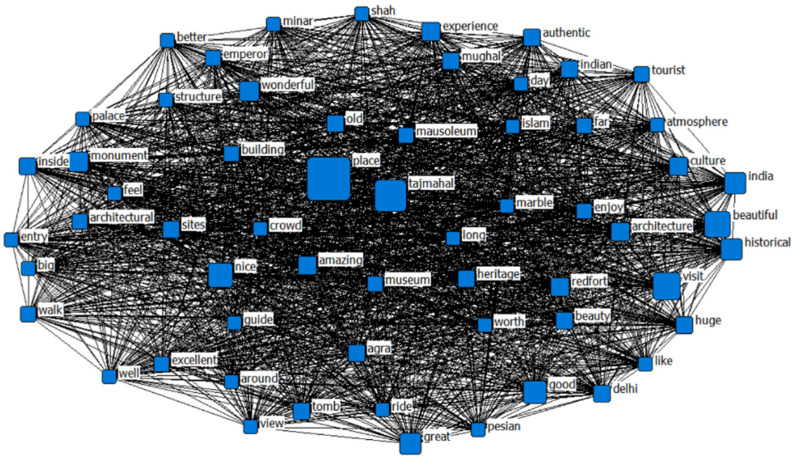
The visibility of the top frequency words in a network.

**Figure 3 behavsci-13-00923-f003:**
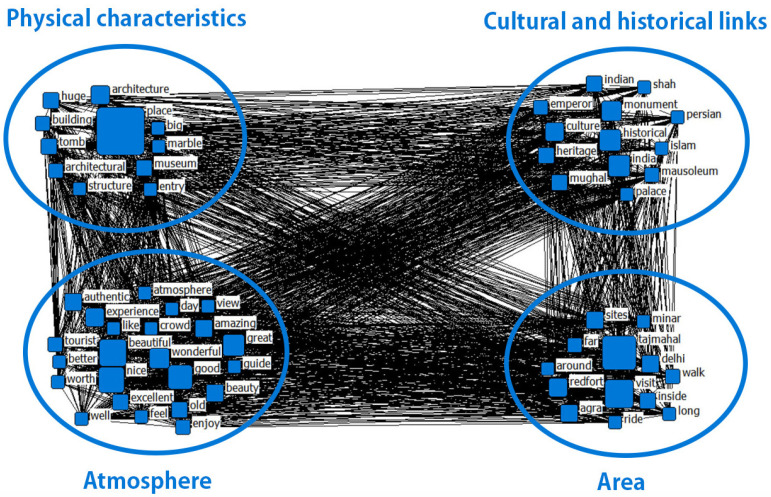
CONCOR analysis visualization.

**Table 1 behavsci-13-00923-t001:** Number of review.

No	Destination	Number of Reviews	Average Satisfaction
1	Taj Mahal	4987	4.2
2	Red Fort	1471	4.5
3	Qutub Minar	1323	4.5
4	Humayun’s Tomb	148	4.5
	Total	7929	
	Average Rating	4.425	

**Table 2 behavsci-13-00923-t002:** Summarization of overall satisfaction rating.

Rating	Frequency	Percent	Cumulative Percent (%)
1	110	1.4%	1.4%
2	110	1.4%	2.8%
3	332	4.2%	7%
4	1773	22.5%	29.5%
5	5604	70.5%	100%
Total	7929	100%	
Average Rating		4.425	

**Table 3 behavsci-13-00923-t003:** The frequency of occurrence for the top 60 most frequently used keywords.

Rank	Words	Freq	Rank	Words	Freq
1	place	2072	31	far	229
2	tajmahal	1312	32	mausoleum	221
3	visit	987	33	tourist	200
4	beautiful	911	34	architectural	198
5	nice	796	35	enjoy	190
6	good	735	36	emperor	179
7	india	606	37	walk	175
8	historical	601	38	building	175
9	great	578	39	persian	156
10	wonderful	474	40	guide	154
11	monument	474	41	long	154
12	experience	443	42	ride	153
13	architecture	431	43	crowd	142
14	redfort	410	44	feel	140
15	culture	403	45	better	137
16	amazing	381	46	islam	136
17	authentic	361	47	worth	135
18	sites	347	48	big	135
19	agra	333	49	minar	134
20	beauty	331	50	like	133
21	delhi	330	51	marble	132
22	old	295	52	shah	130
23	tomb	288	53	entry	130
24	mughal	286	54	structure	126
25	inside	284	55	day	125
26	indian	272	56	view	125
27	huge	269	57	atmosphere	124
28	heritage	268	58	around	123
29	excellent	247	59	well	121
30	museum	241	60	palace	119

**Table 4 behavsci-13-00923-t004:** Frequency and centrality of words in comparison.

Factor	Frequency		Degree		Eigenvector		Factor	Frequency		Degree		Eigenvector	
Word	Freq	Rank	Coef.	Rank	Coef.	Rank	Word	Freq	Rank	Coef.	Rank	Coef.	Rank
place	2072	1	20.9556942	1	20.9556942	1	far	229	31	3.0536248	33	0.08081352	31
tajmahal	1312	2	16.2729339	2	16.2729339	2	mausoleum	221	32	3.0587413	32	0.08060642	33
visit	987	3	15.9178541	3	15.9178541	3	tourist	200	33	3.0784776	31	0.07976196	34
beautiful	911	4	15.5370578	4	15.5370578	4	architectural	198	34	3.0426667	35	0.08061526	32
nice	796	5	12.0258153	5	12.0258153	5	enjoy	190	35	2.6348291	37	0.07767155	35
good	735	6	10.8567545	6	10.8567545	7	emperor	179	36	3.0455855	34	0.07747577	36
india	606	7	8.9988798	8	9.0058767	8	walk	175	37	2.8606791	36	0.07546808	38
historical	601	8	8.3612359	9	8.9988798	6	building	175	38	2.6063255	38	0.07581173	37
great	578	9	8.0874707	7	9.3188524	10	persian	156	39	2.2701199	39	0.07491844	39
wonderful	474	10	9.3188524	10	8.5157984	9	guide	154	40	2.2199784	40	0.07183775	40
monument	474	11	9.0058767	11	8.3612359	12	long	154	41	2.1897251	41	0.06378266	43
experience	443	12	6.4574598	13	7.4846939	14	ride	153	42	2.0910537	43	0.05988075	47
architecture	431	13	7.4846930	12	8.0874707	13	crowd	142	43	1.8498628	45	0.06093764	45
redfort	410	14	6.5011547	14	6.8456178	16	feel	140	44	2.1071373	42	0.05991322	46
culture	403	15	8.5157984	15	6.5011547	11	better	137	45	1.7636112	46	0.06818447	41
amazing	381	16	6.8456178	18	5.2278451	15	islam	136	46	2.0881299	44	0.06678006	42
authentic	361	17	5.9956841	16	6.4574598	17	worth	135	47	1.7307287	47	0.06126443	44
sites	347	18	4.1145127	17	5.9956841	19	big	135	48	1.7044152	48	0.05958288	48
agra	333	19	5.2278451	20	4.4498454	18	minar	134	49	1.6634875	50	0.05767134	50
beauty	331	20	4.2297841	19	4.6398514	20	like	133	50	1.6707956	49	0.05768027	49
delhi	330	21	4.2049121	23	4.1145127	22	marble	132	51	1.5955151	52	0.057661961	51
old	295	22	4.6398514	24	3.9803831	21	shah	130	52	1.5041550	54	0.05697171	52
tomb	288	23	3.4110548	21	4.2297841	24	entry	130	53	1.6159809	51	0.01701071	59
mughal	286	24	4.4498454	22	4.2049121	23	structure	126	54	1.2717595	56	0.05528623	54
inside	284	25	2.8606791	25	3.8794975	25	day	125	55	1.4062186	55	0.05596739	53
indian	272	26	3.5156487	27	3.5156487	26	view	125	56	1.5757815	53	0.05193791	55
huge	269	27	3.8545844	26	3.8545844	28	atmosphere	124	57	1.2541935	57	0.04902190	56
heritage	268	28	3.4203640	28	3.4203664	30	around	123	58	1.0553977	59	0.04179074	58
excellent	247	29	3.8794950	29	3.4110548	27	well	121	59	1.1869553	58	0.04217546	57
museum	241	30	2.2199784	30	3.2283055	29	palace	119	60	0.9662252	60	0.01658740	60

**Table 5 behavsci-13-00923-t005:** CONCOR significant words.

	Extracted Words	Significant Words
Physical characteristic	place/architecture/tomb/museum/architectural/building/huge/big/marble/entry/structure	place/architecture/tomb/ museum/architectural/building/structure
Cultural and historical link	historical/mughal/indian/heritage/emperor/ culture/palace/mausoleum/monument/shah/ persian/islam	historical/mughal/indian/ heritage/emperor/culture/ palace/mausoleum/ monument/shah/persian/islam
Atmosphere	beautiful/nice/good/great/wonderful/ experience/amazing/authentic/beauty/old/ excellent/tourist/enjoy/guide/crowd/feel/ better/worth/like/day/view/atmosphere/well	nice/good/great/wonderful/ experience/authentic/worth/ atmosphere
Area	visit/redfort/india/agra/delhi/inside/far/walk/ long/ride/minar/around/tajmahal/sites	redfort/india/agra/delhi/ minar/tajmahal/sites

**Table 6 behavsci-13-00923-t006:** Factor analysis results.

Factor	Words	Factor Loading	Eigen Value	Variance (%)
Physical characteristic	architecture	0.872	3.330	34.025
	tomb	0.823		
	museum	0.853		
	building	0.866		
Cultural and historical link	historical	0.885	1.612	24.665
	indian	0.805		
	heritage	0.835		
	culture	0.879		
	mausoleum	0.856		
Atmosphere	good	0.878	0.188	17.337
	great	0.780		
	experience	0.801		
	atmosphere	0.858		
Area	tajmahal	0.795	0.870	17.432
	delhi	0.796		
	redfort	0.837		
	india	0.793		
	minar	0.740		
KMO (Kaiser–Meyer–Olkin) = 0.826				
Bartlett chi-square (*p*) = 1225.834 (*p* < 0.001)				

**Table 7 behavsci-13-00923-t007:** Results of linear regression analysis.

Model	Unstandardized Coef.		Standardized Coef.	*p*	*t*
	B	Std. Error	Beta		
(Constant)	3.555	0.105		<0.001	33.819 ***
Physical characteristics	0.340	0.093	0.319	<0.001	3.668 ***
Cultural and historical link	0.147	0.072	0.179	0.043	2.054 *
Atmosphere	0.291	0.081	0.309	0.001	3.581 **
Area	0.119	0.067	0.151	0.081	1.766

Dependent variable: customer satisfaction (CS); R^2^ = 0.430; adjusted R^2^ = 0.406; F = 642.000; * *p* < 0.05, ** *p* < 0.01, *** *p* < 0.001.

## Data Availability

Data are contained within the article.

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
