# Peer review of "Analyzing Online Reviews to Uncover Customer Satisfaction Factors in Indian Cultural Tourism Destinations"

_behavsci, 2023, doi:10.3390/bs13110923_

Round 1

Reviewer 1 Report

Comments and Suggestions for Authors

Heritage marketing is an interesting research area. Please see my comments in the attached document.

Author Response

Review 1

To begin with, we would like to express our sincere appreciation for reviewing and providing feedback on our manuscript. Your valuable insights and constructive comments are instrumental in improving the quality and effectiveness of this manuscript, and we are truly grateful for your contribution.

We kindly request that you take a moment to review the revised version of our manuscript. Your expertise and perspective are highly regarded, and we believe your input will help ensure that the final document is of the highest standard.

  1. Introduction

"In the modern era, word-of mouth is widely regarded as an important tool for attracting and retaining clients." (48) Is reference [6] the source for this statement? As WOM is regarded as an essential tool, I expect to see the concept discussed in more detail in the paper.

Your claim about a "scarcity of research exploring sustainability" (70) requires a reference(s).

The statement at (48) indeed references source [6], regarding the statement on line (70) about the "scarcity of research exploring sustainability," it should have been attributed to reference [7]. The omission was an oversight, and I apologize for not including the reference at that point. We acknowledge the importance of proper citation and will ensure that references are included to support this claim in the revised manuscript.

  1. Literature Review

2.1. Heritage Tourism

Reference to more recent sources would strengthen the discussion about Heritage

Tourism, i.e. including more references to support the discussion, for example:

Apostolakis, A. (2017). The convergence process in heritage tourism. In The Political Nature of Cultural Heritage and Tourism (pp. 1-18). Routledge.

Santa, E. D., & Tiatco, A. (2019). Tourism, heritage, and cultural performance:

Developing a modality of heritage tourism. Tourism Management Perspectives, 31, 301-309.

Timothy, D. J. (2018). Making sense of heritage tourism: Research trends in a maturing field of study. Tourism management perspectives, 25, 177-180.

Murzyn-Kupisz, M. (2012). Cultural, economic and social sustainability of heritage tourism: issues and challenges. Economic and Environmental Studies (E&ES), 12(2), 113-133.

We have revised the manuscript by including more recent sources to enhance the discussion on Heritage Tourism ( line 88-96). Specifically, we have added the following references to our discussion:

Apostolakis, A. (2017). The convergence process in heritage tourism. In The Political Nature of Cultural Heritage and Tourism (pp. 1-18). Routledge.

Murzyn-Kupisz, M. (2012). Cultural, economic and social sustainability of heritage tourism: issues and challenges. Economic and Environmental Studies (E&ES), 12(2), 113-133.

Santa, E. D., & Tiatco, A. (2019). Tourism, heritage, and cultural performance: Developing a modality of heritage tourism. Tourism Management Perspectives, 31, 301-309.

These additions have further enriched our analysis of heritage tourism and its various dimensions. We appreciate the reviewer's insightful recommendation, which has strengthened the quality of our work.

2.2. Online revlews and tourism

Do the first six lines of the first paragraph all refer to reference [17]? You mention user-generated content - a brief definition would be good.

Yes, the first six lines of the first paragraph all refer to reference [17]. We have incorporated additional information about user-generated content, as suggested. Specifically, we have included a brief definition to enhance the reader's understanding of this concept.

The second paragraph can do with a critical discussion i.e. by including the potential weaknesses in WOM, for example:

Schuckert, M., Liu, X. and Law, R., 2015. Hospitality and tourism online reviews: Recent trends and future directions. Journal of Travel & Tourism Marketing, 32(5), pp.608-621.

Sotiriadis, M.D. and Van Zyl, C., 2013. Electronic word-of-mouth and online reviews in tourism services: the use of twitter by tourists. Electronic Commerce Research, 13, pp.103-124.

Hu, X. and Yang, Y., 2021. What makes online reviews helpful in tourism and hospitality? A bare-bones meta-analysis. Journal of hospitality marketing & management, 30(2), pp.139-158.

Reyes-Menendez, A., Saura, I.R. and Filipe, F., 2019. The importance of behavioral data to identify online fake reviews for tourism businesses: A systematic review. Peer)

Computer Science, 5, p.e219.

Xu, H., Lovett, J., Cheung, L.T., Duan, X., Pei, Q. and Liang, D., 2021. Adapting to social media: the influence of online reviews on tourist behaviour at a world heritage site in China. Asia Pacific Journal of Tourism Research, 26(10), pp. 1125-1138.

Wang, C., Liu, S., Zhu, S. and Hou, Z., 2022. Exploring the effect of the knowledge redundancy of online reviews on tourism consumer purchase behaviour: based on the knowledge network perspective. Current Issues in Tourism, pp.1-16.

Zelenka, J., Azubuike, T. and Pásková, M., 2021. Trust model for online reviews of tourism services and evaluation of destinations. Administrative Sciences, 11(2), p.34.

Thank you for your suggestion. In response, we have revised the second paragraph to include a discussion of potential weaknesses in Word-of-Mouth (WOM) (line 150-160). We have incorporated insights from the following references:

Reyes-Menendez, A., Saura, I.R. and Filipe, F., 2019. The importance of behavioral data to identify online fake reviews for tourism businesses: A systematic review. Peer)

Sotiriadis, M.D. and Van Zyl, C., 2013. Electronic word-of-mouth and online reviews in tourism services: the use of twitter by tourists. Electronic Commerce Research, 13, pp.103-124.

Wang, C., Liu, S., Zhu, S. and Hou, Z., 2022. Exploring the effect of the knowledge redundancy of online reviews on tourism consumer purchase behaviour: based on the knowledge network perspective. Current Issues in Tourism, pp.1-16.

Xu, H., Lovett, J., Cheung, L.T., Duan, X., Pei, Q. and Liang, D., 2021. Adapting to social media: the influence of online reviews on tourist behaviour at a world heritage site in China. Asia Pacific Journal of Tourism Research, 26(10), pp. 1125-1138.

Do lines (129) - (133) relate to source (22)?

Yes, it is related to the reference [22].

2.3. Big Data Analytics

Do lines (140) - (143) relate to source (26)?

Yes, it is related to the reference [26].

  1. Methodology

The paper can do with a discussion around the choice of a mixed methods approach. In this regard, you will find the source below very useful.

Walker, C., & Baxter, J. (2019). Method sequence and dominance in mixed methods research: A case study of the social acceptance of wind energy literature. International Journal of Qualitative Methods, 18, 1609406919834379.

Thank you for your valuable suggestion. We have now incorporated a discussion about our choice of a mixed methods approach in the paper. We appreciate the recommended source, Walker and Baxter (2019), which greatly contributed to enhancing our understanding and the rationale behind our selection of a mixed methods approach in this study.

In Table 1, I am unsure how helpful an 'average satisfaction score' is.

The 'average satisfaction score' in Table 1 represents an overall rating derived from respondents' assessments, where 5 is the highest possible rating. It serves as a condensed and easily interpretable measure of the general satisfaction level among respondents.

  1. Result

4.1. Data Pre-Processing

I find it challenging to make sense of words in Table 3. such as 'place', 'visit', 'nice' 'sites',

'old', to name a few.

Finally, I do not feel that the analysis gives the reader an idea of what did and didn't work for the visitors. It would be difficult to make sense of the customer experience.

Table 3 lists the top 60 most frequently used words in the dataset. For instance, the word 'place' is the most commonly used word, occurring 2,072 times. These word frequencies are intended to provide insights into the most prevalent terms used in customer feedback.

Regarding the analysis of what did and didn't work for visitors, this information can be found in Table 7. In Table 7, we present the results of our linear regression analysis, which helps in understanding the factors influencing customer satisfaction. This table provides a more detailed and in-depth examination of the customer experience, highlighting what factors positively or negatively affect satisfaction levels.

  1. Discussion and Conclusion

You demonstrate good knowledge of data analysis tools; however, I am not sure the paper provides management of heritage destinations with critical factors that influence customer satisfaction/customer experience. Using a mixed methods approach to data analysis may not have been the best choice.

We revised and divided the section to

  1. Discussion and Implications-5.1 Discussion, 5.2 Implications
  2. Conclusion-6.1 Conclusion, 6.2 6.2 Limitations and Future Research

We added the theoretical and practical implications in lines 427-447. In addition, we also provided the limitations and suggestions for future research in lines 463-470.

Reviewer 2 Report

Comments and Suggestions for Authors

The primary purpose of this review is to give some pointers to the paper’s authors entitled “Exploring key attributes influencing customer satisfaction in Indian heritage tourism destinations: an online review analysis”. Overall, my impression is that the paper might be ready to be published if the authors strongly revise it. The current version shows several strengths. First, the introduction addresses the subject matter by indicating the tangible and no tangible advantages of heritage tourism for destinations (lines 30-49). Similarly, the authors spot the research gap (lines 60-63) and set out the general research objective (lines 66-67). Second, the authors indicate the employed software to gather information in the methodology (lines 168-186). Likewise, they account for the fieldwork context (lines 187-195). Equally, they display information about the sample profile in Table 1. Third, it is hard to fault the thread and sequence of statistical analyses from descriptive, semantic and cluster tests to causal linear regression tests in the result section. It is generally accepted that the proper sequence should start with descriptive tests and finish with causal tests. Therefore, it is all right.

(1)    However, the current version of the paper shows severe shortcomings as follows:

(2)    (1)   I do not see the point of dealing with tourist motivations to visit heritage sites because it is misguided and side-tracked (lines 50-59). In other words, it is a digression as the assumed paper topic is not visitors’ motivation but determining satisfaction by analysing the online reviews. Therefore, let me suggest that you delete it.

(3)    (2)   It is disgraceful of the authors to reiterate the exact text twice in the introduction (lines 69-79). Please delete it.

(4)    (3)   It is advisable to append an advance of the paper structure to the introduction. Please make a paragraph to describe the paper structure at the end of the introduction.

(5)   It is thoughtless of the authors to fail to find ground for hypotheses in the literature review. In other words, the literature review's current version is nothing but a little introductory content such as heritage tourism, online reviews and big data analytics. It is introductory because it defines it but does not support any hypothesis or theoretical proposition. Please move this content to the introduction so that all of this can be presented in one paragraph.

(6)   Reviewing the literature and analysing the results should be aligned. As seen in the analysis of the results section, four are the determining factors stemming from the user-generated content, such as physical characteristics, atmosphere, cultural and historical links and area. Therefore, let me recommend that the authors put forward four different hypotheses as follows:

a.       H1: The physical characteristics of the heritage sites determine tourist satisfaction

b.      H2: The atmosphere of the heritage sites determines tourist satisfaction

c.       H3: The cultural and historical links of the heritage sites determine tourist satisfaction

d.      H4: The area of the heritage sites determines tourist satisfaction.

(7)   The review of the literature section should find ground for supporting these hypotheses and, consequently, be divided into these four subsections. In these subsections, the authors should describe the variables belonging to these factors so that they justify these factors' ingredients.

(8)   The methodology does not ensure the validity and reliability of the measuring instruments. Please describe your measuring instruments and disclose how valid and reliable they are.

(9)    It is worth tackling with the statistical analysis tools in the methodology. Therefore, I wonder if the authors might dedicate a paragraph to describe their statistical tools in the methodology section.

(10) Table 7 should refer to the hypotheses. Equally, the empirical contrast of the four hypotheses should be developed at the end of the result section.

(11)Discussing and concluding are two different tasks. While the former should compare the obtained evidence and other papers’ evidence with the aim of gaining insight into the obtained results, the latter should summarise, highlight the theoretical contribution, come up with practical implications, acknowledge limitations and put forward future lines of research. On this basis, let me suggest that you distinguish both by creating two different sections. In addition to creating these new sections, a wide range of measures should be taken as follows:

a.       Lines 376-386 should be moved to the new discussion section. Moreover, it should be more insightful by delving into the results and come up with sharp interpretations.

b.       As lines 387-395 are a kind of summary, it is suitable for prefacing the new conclusion section.

c.       I do not think it is correct to display numbers in the conclusion section (396-399). Please, delete it or revise it so that it summarises the results.

d.       Lines 400-408 are practical implications. Nevertheless, it is too general and needs to be more specific to the obtained results.

e.       Please, acknowledge limitations and put forward future lines of research at the end of the conclusion section.

I hope these comments help the authors and encourage them to move forward

Author Response

Review 2

To begin with, we would like to express our sincere appreciation for reviewing and providing feedback on our manuscript. Your valuable insights and constructive comments are instrumental in improving the quality and effectiveness of this manuscript, and we are truly grateful for your contribution.

We kindly request that you take a moment to review the revised version of our manuscript. Your expertise and perspective are highly regarded, and we believe your input will help ensure that the final document is of the highest standard.

The primary purpose of this review is to give some pointers to the paper’s authors entitled “Exploring key attributes influencing customer satisfaction in Indian heritage tourism destinations: an online review analysis”. Overall, my impression is that the paper might be ready to be published if the authors strongly revise it. The current version shows several strengths. First, the introduction addresses the subject matter by indicating the tangible and no tangible advantages of heritage tourism for destinations (lines 30-49). Similarly, the authors spot the research gap (lines 60-63) and set out the general research objective (lines 66-67). Second, the authors indicate the employed software to gather information in the methodology (lines 168-186). Likewise, they account for the fieldwork context (lines 187-195). Equally, they display information about the sample profile in Table 1. Third, it is hard to fault the thread and sequence of statistical analyses from descriptive, semantic and cluster tests to causal linear regression tests in the result section. It is generally accepted that the proper sequence should start with descriptive tests and finish with causal tests. Therefore, it is all right.

However, the current version of the paper shows severe shortcomings as follows:

  1.    I do not see the point of dealing with tourist motivations to visit heritage sites because it is misguided and side-tracked (lines 50-59). In other words, it is a digression as the assumed paper topic is not visitors’ motivation but determining satisfaction by analysing the online reviews. Therefore, let me suggest that you delete it.
  2. It is disgraceful of the authors to reiterate the exact text twice in the introduction (lines 69-79). Please delete it.

Thank you for checking. We deleted the text in lines 69-79.

  1. It is advisable to append an advance of the paper structure to the introduction. Please make a paragraph to describe the paper structure at the end of the introduction.

      We added the structure explanation at the end of the introduction in line 64-68:

“This study was organized into several sections. Section 2 presents previous studies and research papers on heritage tourism, online review, and big data analytics. Section 3 outlines the research methodology and Section 4 provides analytical results with explanatory tables. Lastly, Section 5 includes a discussion of the findings, study limitations, and suggestions for future research.”

  1. It is thoughtless of the authors to fail to find ground for hypotheses in the literature review. In other words, the literature review's current version is nothing but a little introductory content such as heritage tourism, online reviews and big data analytics. It is introductory because it defines it but does not support any hypothesis or theoretical proposition. Please move this content to the introduction so that all of this can be presented in one paragraph.

The review of the literature section should find ground for supporting these hypotheses and, consequently, be divided into these four subsections. In these subsections, the authors should describe the variables belonging to these factors so that they justify these factors' ingredients.

      This study created the factors based on the CONCOR analysis result. Therefore, it is not possible to have the hypotheses in the literature part.  We provide the previous study that use a similar methodology for a better understanding.

Handani, N. D., & Kim, H. S. (2023). Unlocking customer satisfaction of Halal restaurant in South Korea through online review analysis. Environment and Social Psychology, 7(2).

  1. Reviewing the literature and analysing the results should be aligned. As seen in the analysis of the results section, four are the determining factors stemming from the user-generated content, such as physical characteristics, atmosphere, cultural and historical links and area. Therefore, let me recommend that the authors put forward four different hypotheses as follows:
  2. H1: The physical characteristics of the heritage sites determine tourist satisfaction
  3. H2: The atmosphere of the heritage sites determines tourist satisfaction
  4. H3: The cultural and historical links of the heritage sites determine tourist satisfaction
  5. H4: The area of the heritage sites determines tourist satisfaction.

As mentioned above, it is difficult to have the hypotheses. Therefore, we added the explanation in the methodology part in line 203-208:

“Based on the factors from the CONCOR analysis, this study employed quantitative analysis methods. Factor analysis and linear regression analysis are conducted with the SPSS 26, to ascertain the factors reflecting customer experience and their correlation with customer satisfaction. The analysis involved scrutinizing customer satisfaction rating scales, which ranged from 1 to 5, with 1 star representing dissatisfied customers and 5 stars representing satisfied customers. The flow of the research is as shown in Figure 1.”

  1. The methodology does not ensure the validity and reliability of the measuring instruments. Please describe your measuring instruments and disclose how valid and reliable they are.

  1. It is worth tackling with the statistical analysis tools in the methodology. Therefore, I wonder if the authors might dedicate a paragraph to describe their statistical tools in the methodology section.

      We added the research flow to show the flow of the research and mentioned the analysis tool software. It can be seen in Figure 1.

  1. Table 7 should refer to the hypotheses. Equally, the empirical contrast of the four hypotheses should be developed at the end of the result section.

      Since we don’t have the hypotheses for this research, it is difficult to mention them in the results part.

  1. Discussing and concluding are two different tasks. While the former should compare the obtained evidence and other papers’ evidence with the aim of gaining insight into the obtained results, the latter should summarise, highlight the theoretical contribution, come up with practical implications, acknowledge limitations and put forward future lines of research. On this basis, let me suggest that you distinguish both by creating two different sections. In addition to creating these new sections, a wide range of measures should be taken as follows:
  2. Lines 376-386 should be moved to the new discussion section. Moreover, it should be more insightful by delving into the results and come up with sharp interpretations.
  3. As lines 387-395 are a kind of summary, it is suitable for prefacing the new conclusion section.
  4. I do not think it is correct to display numbers in the conclusion section (396-399). Please, delete it or revise it so that it summarises the results.
  5. Lines 400-408 are practical implications. Nevertheless, it is too general and needs to be more specific to the obtained results.
  6. Please, acknowledge limitations and put forward future lines of research at the end of the conclusion section.

I hope these comments help the authors and encourage them to move forward

We divided the section:

  1. Discussion and Implications-5.1 Discussion, 5.2 Implications
  2. Conclusion-6.1 Conclusion, 6.2 6.2 Limitations and Future Research

We added the theoretical and practical implications in lines 427-447. In addition, we also provided the limitations and suggestions for future research in lines 463-470.

Reviewer 3 Report

Comments and Suggestions for Authors

Dear Authors,

I find this manuscript interesting as it focuses on a complex analysis of customer satisfaction in relation with heritage tourism in India using both quantitative  and qualitative methods.

Keywords: you should avoid using key words which are also in the title: consumer satisfaction

Introduction:

At the end of introduction between rows 60-79 some sentences are written twice.

At the end of the introduction, you should formulate the research questions in relation with the aim of the study.

2. Literature Review

2.1. Heritage Tourism

For the phases between rows 88 and 93 the authors should provide at least 3 or 4 other new citations to validate the fact that heritage tourism sector is highly profitable: ʺThe heritage tourism sector is highly profitable, with tourists spending more on their destinations than other types of tourism, and staying longer than regular tourists. There is evidence that heritage tourism is profitable, as heritage tourists tend to be well-educated, older, responsible, cosmopolitan, generous in spending, stay longer, require high-quality services, and are interested in learning more about unique and authentic culture [12].

For the following phase (rows 98-99) citation is required: ʺBy 2033, the market is projected to reach a total value of US$1,316.4 billion, with a compound annual growth rate of 7.2%ʺ.

For the next afirmation: ʺA great deal of this growth can be attributed to the surging popularity of heritage tourism around the world, with India also gaining considerable traction within this sectorʺ, can the authors provide several examples to support the statement that India is registering an increase in heritage tourism? Or it is a statemen taken from the specialized literature? In this case it needs citation

2.3. Big Data Analytics

In the following sentence you need to add 2 or 3 citations (rows 138-140):  ʺIn recent years, scientists have been able to gather valuable insights from vast amounts of unstructured information online as a result of technological advancesʺ [26].

4. Discussion and conclusions

In the discussion section several details should be added:

(1)  You should present critically the results synthetically refering to the expected benefits and contribution of the study (methodologically and  implication for policy, practice);

(2)  the limitations of the study and future research.

Minor comments:

Mispelling words: cooccurrence matrix (p. 4, row 163).

The first phrase of the article is too large. It can be divided into two sentences:

The tourism industry has experienced an intense level of competition in recent years  30 and recent studies on heritage tourism have been refocused from product-centric methodologies to one that focuses on visitors, this new approach aims to investigate the inter- 32 actions between tourists and heritage sites in the context of this increased competition in  33 tourism [1].

Author Response

Review 3

To begin with, we would like to express our sincere appreciation for reviewing and providing feedback on our manuscript. Your valuable insights and constructive comments are instrumental in improving the quality and effectiveness of this manuscript, and we are truly grateful for your contribution.

We kindly request that you take a moment to review the revised version of our manuscript. Your expertise and perspective are highly regarded, and we believe your input will help ensure that the final document is of the highest standard.

Dear Authors,

I find this manuscript interesting as it focuses on a complex analysis of customer satisfaction in relation with heritage tourism in India using both quantitative  and qualitative methods.

  1. Keywords: you should avoid using key words which are also in the title: consumer satisfaction

We changed it to the customer experience.

Introduction:

  1. At the end of introduction between rows 60-79 some sentences are written twice.

We deleted the repeated part.

  1. At the end of the introduction, you should formulate the research questions in relation with the aim of the study.
  2. (Heritage Tourism) For the phases between rows 88 and 93 the authors should provide at least 3 or 4 other new citations to validate the fact that heritage tourism sector is highly profitable: ʺThe heritage tourism sector is highly profitable, with tourists spending more on their destinations than other types of tourism, and staying longer than regular tourists. There is evidence that heritage tourism is profitable, as heritage tourists tend to be well-educated, older, responsible, cosmopolitan, generous in spending, stay longer, require high-quality services, and are interested in learning more about unique and authentic culture [12].

We added one more paragraph about the heritage tourism with some citations in lines 77-84 : Heritage tourism provides a multifaceted and multidisciplinary perspective on this developing field. It encompasses the fusion of cultural and political influences, highlights the significance of cultural performance, and charts the evolution of re-search trends within heritage tourism [12]. Furthermore, it accentuates the critical as-pect of upholding cultural, economic, and social sustainability [13]. The essential role of cultural performance in heritage tourism is rooted in its ability to bring heritage sites to life, establish connections with the past, educate and engage visitors, and create enduring memories, all while preserving and promoting cultural traditions [14].

For the following phase (rows 98-99) citation is required: ʺBy 2033, the market is projected to reach a total value of US$1,316.4 billion, with a compound annual growth rate of 7.2%ʺ.

We have relocated the citation [18] to the end of the sentence for proper referencing.

  1. For the next afirmation: ʺA great deal of this growth can be attributed to the surging popularity of heritage tourism around the world, with India also gaining considerable traction within this sectorʺ, can the authors provide several examples to support the statement that India is registering an increase in heritage tourism? Or it is a statemen taken from the specialized literature? In this case it needs citation

We added the citation about the india heritage tourism.

Piramanayagam, S., Rathore, S., & Seal, P. P. (2021). Destination image, visitor experience, and behavioural intention at heritage centre. In Tourism in India (pp. 35-52). Routledge.

  1. (Big Data Analytics) In the following sentence you need to add 2 or 3 citations (rows 138-140):  ʺIn recent years, scientists have been able to gather valuable insights from vast amounts of unstructured information online as a result of technological advancesʺ [26].

We added more references to it.

Handani, N. D., & Kim, H. S. (2023). Unlocking customer satisfaction of Halal restaurant in South Korea through online review analysis. Environment and Social Psychology, 7(2).

Kim, Y. J., & Kim, H. S. (2022). The impact of hotel customer experience on customer satisfaction through online reviews. Sustainability14(2), 848.

In the discussion section several details should be added:

  1. You should present critically the results synthetically refering to the expected benefits and contribution of the study(methodologically and  implication for policy, practice);

We divided the section:

  1. Discussion and Implications-5.1 Discussion, 5.2 Implications
  2. Conclusion-6.1 Conclusion, 6.2 6.2 Limitations and Future Research

We added the theoretical and practical implications in lines 427-447.

  1. the limitations of the study and future research.

We added the limitations and suggestions for future research in lines 463-470:

“This study comes with several limitations and these limitations also offer valuable insights for future research directions. The online reviews were sourced from Google Maps, which is the largest search engine globally. However, it is plausible that relying on a single online platform may not encompass all customer preferences. Therefore, for enhanced representativeness, future studies should encompass analyses from various websites. Furthermore, it's worth noting that this study specifically collected data from four heritage destinations. Subsequent research endeavors should aim to encompass a broader array of heritage tourism destinations for a more comprehensive understanding.”

Minor comments:

  1. Mispelling words: cooccurrence matrix (p. 4, row 163).

We revised it into co-occurrence.

  1. The first phrase of the article is too large. It can be divided into two sentences:

The tourism industry has experienced an intense level of competition in recent years  30 and recent studies on heritage tourism have been refocused from product-centric methodologies to one that focuses on visitors, this new approach aims to investigate the inter- 32 actions between tourists and heritage sites in the context of this increased competition in  33 tourism [1].

We divided it into 2 sentences.

Round 2

Reviewer 1 Report

Comments and Suggestions for Authors

The additions strengthened the manuscript. 

Author Response

Thank you for your valuable feedback on our revised paper, "Analyzing Online Reviews to Uncover Customer Satisfaction Factors in Indian Cultural Tourism Destinations." Your comments were instrumental in refining our work. We've addressed your concerns by enhancing our literature review, adding hypotheses, and improving the discussion and conclusion sections. Your guidance has significantly improved our paper, and we're committed to further revisions to meet academic standards.

We genuinely appreciate your guidance, which has undoubtedly helped us enhance the quality and clarity of our paper. Your feedback has encouraged us to move forward with a more robust and coherent research contribution. We are committed to addressing these issues in our next revision and ensuring that the paper meets the highest standards of academic quality.

Thank you

Reviewer 2 Report

Comments and Suggestions for Authors

This second review outlines how the authors have revised their paper, “Analysing Online Reviews to Uncover Customer Satisfaction Factors in Indian Cultural Tourism Destinations. " I congratulate the authors on having removed some digressions and reiterations from the introduction and successfully appended a description of the paper structure.

Nevertheless, I must insist on several remaining shortcomings as follows:

(1)   The literature review section is too basic and shallow. Defining heritage tourism needs to be more innovative (lines 71-77), and it needs to be groundbreaking. Moreover, justifying this sector's tangible and non-tangible benefits is introductory (lines 78-101). Similarly, not only does touching upon the topic of online reviews and tourism to highlight its importance not contribute significantly, but it also needs to be in line with the obtained empirical evidence shown in the analysis of the result section (lines 102-148). The easiest way to connect the literature review with the analysis of the result section is to put forward hypotheses. Finally, bringing up the issue of Data Analytics in the literature review section seems like it needs to be put in the right place (102-148) insofar as it is more suitable for the methodological section. That said, this specific content is not more methodological than introductory; hence, it is more feasible for initial and starting paragraphs. Please pay careful thought to my previous recommendation regarding the hypotheses. Let me bring your attention to the fact that your regression is perfectly aligned with the hypotheses that I put forward for you. What is only needed to do is to thorough the literature to find theoretical ground for them. Otherwise, your presumed literature review is only basic and general content with no significant contribution.

(1)  The authors can win plaudits for separating the discussion and conclusion sections. Nevertheless, the content distribution is disorganised. To overcome this, a range of measures should be taken as follows:

a.      The actual discussion section is only one paragraph (lines 403-413); the rest is a summary (lines 414-428). Please move this summary to the conclusion section and make more effort to create a more extended discussion section.

b. Discussing the obtained results differs from analysing them (lines 426-427). Please remove the numerical information.

c.      Discussing is more than presenting the obtained results and pointing out they are supported by previous research (lines 429-440). We need to delve into the obtained results by comparing similarities and differences. Please gain insight into similarities and differences by reflecting deeper and gaining further understanding. Once you do so, you can leave this content in the discussion section.

d.     Highlighting the paper's genuine contribution is worth doing, but it is more suitable for the conclusion section than the discussion section (lines 441-451). Please move it to the conclusion section.

e. The discussion and conclusion sections do not need to be divided into subheadings. Using the exact words for the heading and subheading is obvious. Please delete the subheadings within these sections.

I hope these comments help improve the paper and encourage the authors to move forward.

Round 3

Reviewer 2 Report

Comments and Suggestions for Authors

This is the third review of the paper and, unfortunately, the main shortcomings remain. It is true up to the point that the authors have deleted the reiterated text in the introduction and appended a paragraph to describe the paper's structure to the introduction. Moreover, they seem to adopt the hypotheses that I proposed to them and placed them at the end of the literature review section. However, they overlook my previous main suggestions. As a result, there are digressions in the introduction; the literature review section is too basic, obvious and shallow, along with other problems, such as that they muddle through the methodology, the discussion and the conclusion sections. To be specific, let me reiterate some of the pointers and explain further as follows:

(1)   I do not see the point of dealing with tourists’ motivations in the introduction, as the paper’s objective is determining factors by analysing online reviews (lines 53-57). Please delete it.

(2)   The review of the literature is full of basic, obvious and shallow ideas regarding heritage tourism, online review and tourism and big data analytics. Not only is this nothing other than introductory content, it does not make any contribution. Moreover, it makes no sense to put forward hypotheses without any theoretical ground. Please thoroughly review the literature on physical, atmosphere, culture and area of heritage to support the hypotheses. Each hypothesis should be placed after its corresponding supporting theoretical content. Please thoroughly review the literature, aligning the hypotheses and the literature review.

(3)   The methodology needs to be revised. First, it does not ensure the validity nor the reliability of the measuring instruments because it does not even describe the measuring instruments. Second, it is a contradiction in terms when the authors speak of correlations, but what they carry out is a factor analysis and a regression test. Finally, I do not see the point of conducting linear regression analysis when the research objectives are to identify the factors (lines 442-449). Please describe the measuring instruments, delete this contradiction and enrich your research objectives.

(4)   There must be two different sections, such as the discussion and conclusions. These sections do not need to be divided into several subheadings because they are short and this is the accepted way. Finally, these sections should include their appropriate content, which should be relevant. To achieve it, the following actions should be taken:

a.       Setting out the research objectives at the end of the paper (lines 442-449) makes no sense; it should be in the introduction. Please move it to the introduction or delete it.

b.      Discussing means gaining insight into the obtained result by comparing the obtained evidence with other authors’ evidence. Therefore, it is not enough to mention other papers (lines 415-441). Instead, you must delve into your results by indicating similarities and differences between your results and those of other papers. Please develop a proper discussion section.

c.       You know how to discuss (lines 450-461), but you do not know how to place it correctly. Please move it to the discussion section. 

d.      There are two tasks (lines 462-471): discussing and developing practical implications. Please move the practical implications content to the conclusion section.

e.       Do not distinguish subheadings within the conclusion and discussion sections. Please delete the subheadings in both sections.

I hope my reiterate comments make it clearer and the authors move forward
